# Iterative temporal differencing with fixed random feedback alignment support spike-time dependent plasticity in vanilla backpropagation for deep learning

## Abstract

In vanilla backpropagation (VBP), activation function matters considerably in terms of non-linearity and differentiability. Vanishing gradient has been an important problem related to the bad choice of activation function in deep learning (DL). This work shows that a differentiable activation function is not necessary any more for error backpropagation. The derivative of the activation function can be replaced by an iterative temporal differencing (ITD) using fixed random feedback weight alignment (FBA). Using FBA with ITD, we can transform the VBP into a more biologically plausible approach for learning deep neural network architectures. We don't claim that ITD works completely the same as the spike-time dependent plasticity (STDP) in our brain but this work can be a step toward the integration of STDP-based error backpropagation in deep learning.

## 1 Introduction

VBP was proposed around 1987 Rumelhart et al. (1985). Almost at the same time, biologically-inspired convolutional networks was also introduced as well using VBP LeCun et al. (1989). Deep learning (DL) was introduced as an approach to learn deep neural network architecture using VBP LeCun et al. (1989; 2015); Krizhevsky et al. (2012). Extremely deep networks learning reached 152 layers of representation with residual and highway networks He et al. (2016); Srivastava et al. (2015). Deep reinforcement learning was successfully implemented and applied which was mimicking the dopamine effect in our brain for self-supervised and unsupervised learning Silver et al. (2016); Mnih et al. (2015; 2013). Hierarchical convolutional neural network have been biologically inspired by our visual cortex Hubel & Wiesel (1959); Fukushima (1988; 1975); Yamins & DiCarlo (2016).

Geoff Hinton in 1988 proposed recirculation in VBP Hinton & McClelland (1988) which does not require the derivative of the activation function. The recirculation-based backprop is the main inspiration behind our work, an iterative temporal differencing in VBP. He gave a lecture about this approach again in NIPS 2007 Hinton (2007), and recently gave a similar lecture in Standford in 2014 and 2017 to reject the four arguments against the biological foundation of backprop. In his latest related lecture in Standford, he explains the main four arguments by neuroscientists on why VBP is not biologically or neurologically feasible 1.

| Neuroscientist arguments against VBP | Hinton's counter-arguments |
|---|---|
| Unsupervised learning using the Dopamine effect in the brain (reinforcement learning) | Autoencoders (AE) and generative adversarial networks (GAN) |
| Spike instead of sending and receiving real values | Dropout Srivastava et al. (2014) using Bernoulli, Gaussian, and Poisson distribution |
| STDP (**Our core focus and contribution**) as a temporal differencing approach | recirculationHinton & McClelland (1988) |
| Symmetry or symmetrical forward and backward path using symmetrical weights | FBA Lillicrap et al. (2016) |

Table 1: The problems with with artificial neural networks compared to the biological neural networks (brain) according to neuroscientist.

The discovery of fixed random synaptic feedback weights alignments (FBA) in error backpropagation for deep learning started a new quest of finding the biological version of VBP Lillicrap et al. (2016) since it solves the symmetrical synaptic weights problem in backprop. Recently, spike-time dependent plasticity was the important issue with backprop. One of the works in this direction, highly inspired from Hinton's recirculation idea Hinton & McClelland (1988), is deep learning using segregated dendrites Guergiuev et al. (2016). Apical dendrites as the segregated synaptic feedback are claimed to be capable of modeling STDP into the backprop successfully Guergiuev et al. (2016).

ITERATIVE TEMPORAL DIFFERENCING

In this section, we visually demonstrate the ITD using FBA in VBP 1. In this figure, VBP, VBP with FBA, and ITD using FBA for VBP are shown all in one figure. The choice of activation function for this implementation was Tanh function. The ITD was applied to MNIST standard dataset. VBP, FBA, and ITD were compared using maximum cross entropy (MCE) as the loss function 2. Also, ITD with MCE as loss function is compared to ITD with least squared error (LSE) 3.

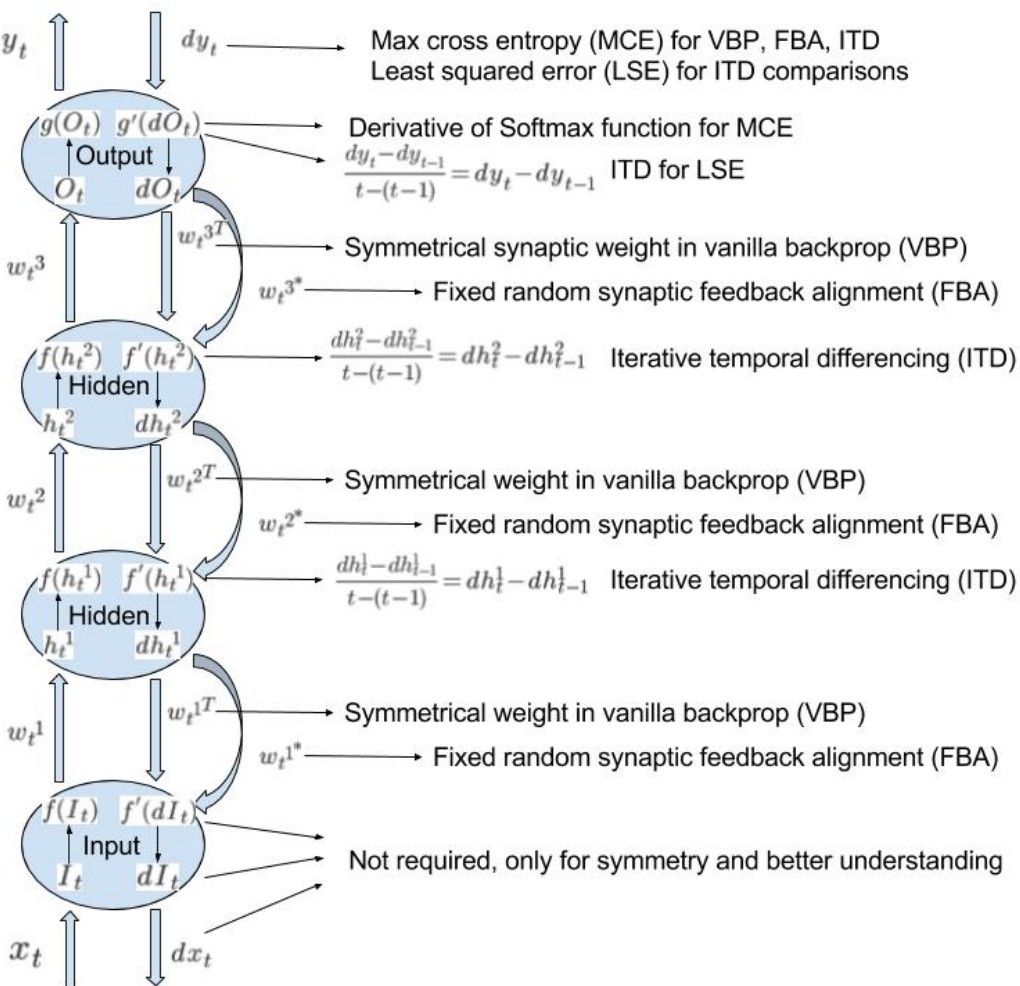

Figure 1: VBP vs FBA vs ITD are all visualized in a 2-layer deep neural network.

The hyper parameters for both of the experiments are equal as follows: 5000 number of iterations/epochs, 0.01 (1e-2) learning rate, 100 minibatch size with shuffling for stochasticity, vanilla stochastic gradient descent is used, 32 for number of hidden layers, 2-layer deep networks. Feed-forward neural network is used as the architecture.

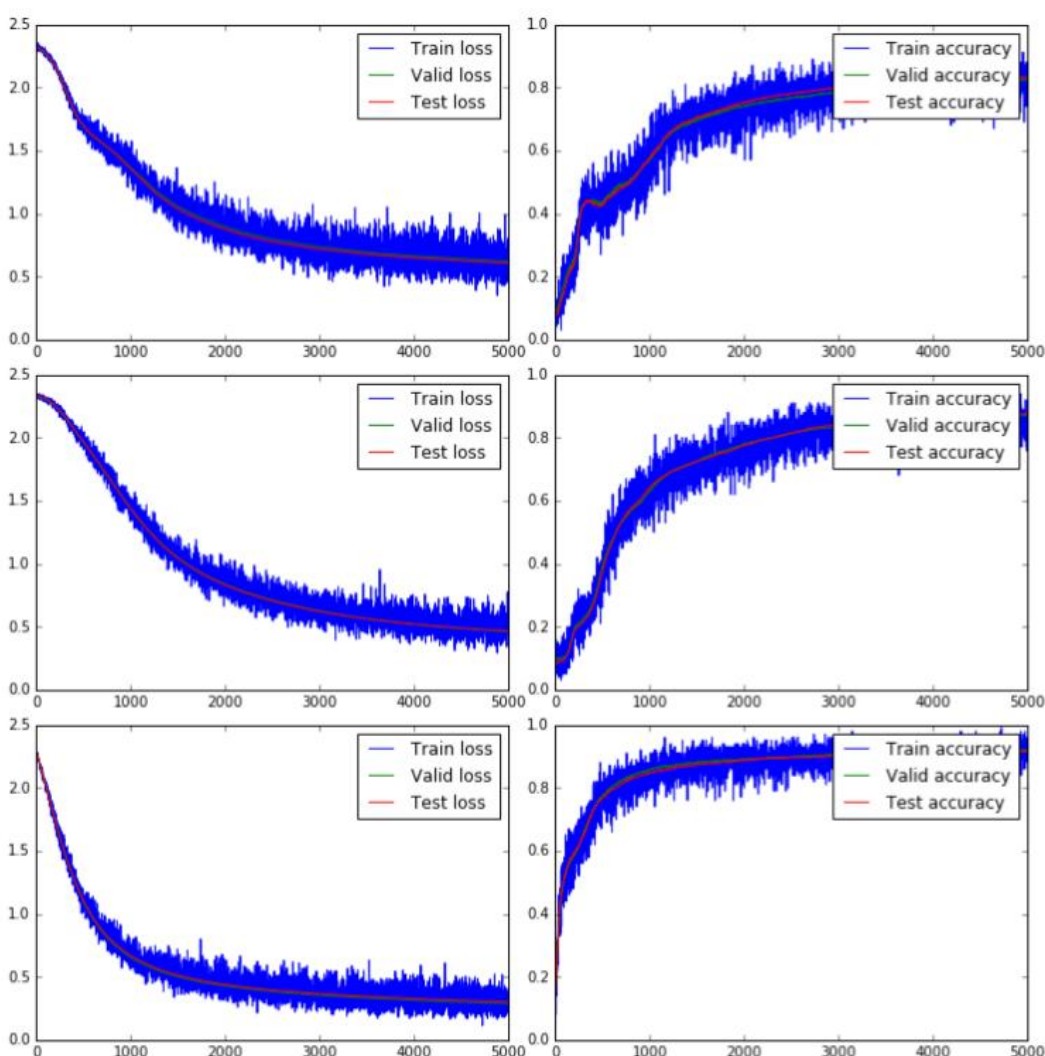

Figure 2: The experimental results on MNIST dataset: (top row) ITD, (middle row) FBA, (bottom row) VBP.

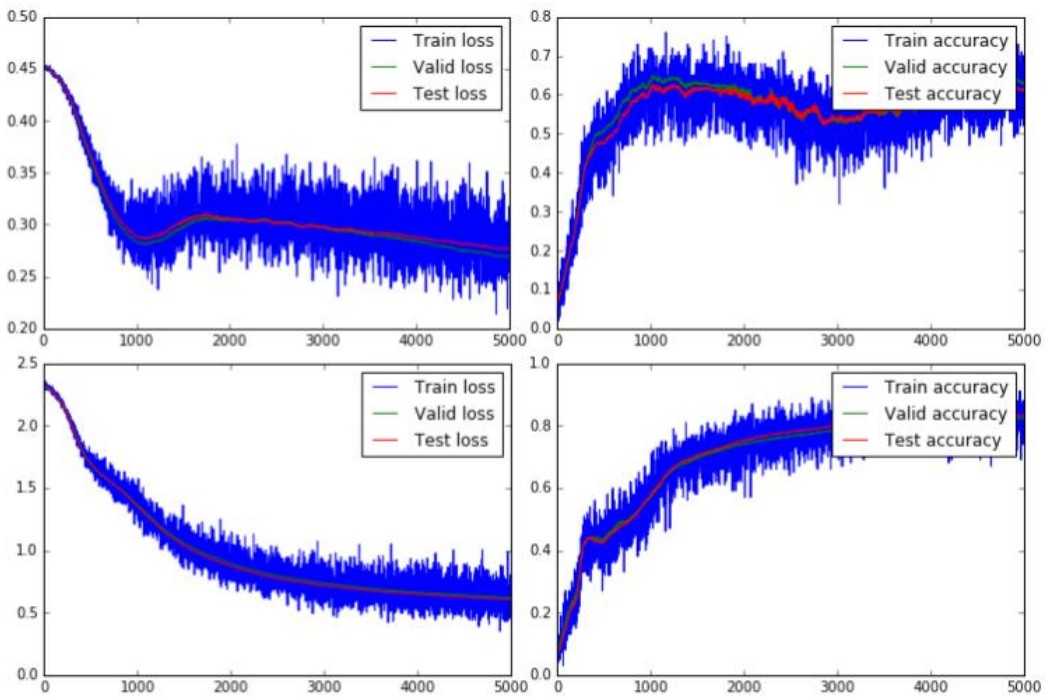

Figure 3: The experimental results on MNIST dataset using ITD with different loss function: (top row) LSE, (bottom row) MCE.

## DISCUSSION & FUTURE VIEW

In this paper, we took one more step toward a more biologically plausible backpropagation for deep learning. After hierarchical convolutional neural network and fixed random synaptic feedback alignment, we believe iterative temporal differencing is a way toward integrating STDP learning process in the brain. We believe the next steps should be to investigate more into the STDP processes details in learning, dopamine-based unsupervised learning, and generating Poisson-based spikes.

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
