# OpenReview forum: "Iterative temporal differencing with fixed random feedback alignment support spike-time dependent plasticity in vanilla backpropagation for deep learning"
_ICLR.cc/2018/Conference — Reject_

### Official Review · AnonReviewer2 · 2017-11-25
**Not clear what exactly the authors want to achieve**

**Rating:** 3
**Confidence:** 4

**Review:**

- This paper is not well written and incomplete. There is no clear explanation of what exactly the authors want to achieve in the paper, what exactly is their approach/contribution, experimental setup, and analysis of their results.

- The paper is hard to read due to many abbreviations, e.g., the last paragraph in page 2.

- The format is inconsistent. Section 1 is numbered, but not the other sections.

- in page 2, what do the numbers mean at the end of each sentence? Probably the figures?

- in page 2, "in this figure": which figure is this referring to?


Comments on prior work:

p 1: authors write: "vanilla backpropagation (VBP)" "was proposed around 1987 Rumelhart et al. (1985)."

Not true. A main problem with the 1985 paper is that it does not cite the inventors of backpropagation. The VBP that everybody is using now is the one published by  Linnainmaa in 1970, extending Kelley's work of 1960. The first to publish the application of VBP to NNs was Werbos in 1982. Please correct.

p 1: authors write: "Almost at the same time, biologically inspired convolutional networks was also introduced as well using VBP LeCun et al. (1989)."

Here one must cite the person who really invented this biologically inspired convolutional architecture (but did not apply backprop to it): Fukushima (1979). He is cited later, but in a misleading way. Please correct.

p 1: authors write: "Deep learning (DL) was introduced as an approach to learn deep neural network architecture using VBP LeCun et al. (1989; 2015); Krizhevsky et al. (2012)."

Not true. Deep Learning was introduced by Ivakhnenko and Lapa in 1965: the first working method for learning in multilayer perceptrons of arbitrary depth. Please correct. (The term "deep learning" was introduced to ML in 1986 by Dechter for something else.)

p1: authors write: "Extremely deep networks learning reached 152 layers of representation with residual and highway networks He et al. (2016); Srivastava et al. (2015)."

Highway networks were published half a year earlier than resnets, and reached many hundreds of layers before resnets. Please correct.


General recommendation: Clear rejection for now. But perhaps the author want to resubmit this to another conference, taking into account the reviewer comments.

---

### Official Review · AnonReviewer1 · 2017-11-26
**Not up to a professional standard.**

**Rating:** 2
**Confidence:** 5

**Review:**

The paper falls far short of the standard expected of an ICLR submission.

The paper has little to no content. There are large sections of blank page throughout. The algorithm, iterative temporal differencing, is introduced in a figure -- there is no formal description. The experiments are only performed on MNIST. The subfigures are not labeled. The paper over-uses acronyms; sentences like “In this figure, VBP, VBP with FBA, and ITD using FBA for VBP…” are painful to read.

---

### Official Review · AnonReviewer3 · 2017-11-27
**The paper claims to work towards a more biological version of error-backpropagation.**

**Rating:** 2
**Confidence:** 5

**Review:**

The paper is incomplete and nowhere near finished, it should have been withdrawn.

The theoretical results are presented in a bitmap figure and only referred to in the text (not explained), and  the results on datasets are not explained either (and pretty bad). A waste of my time.

---

### Decision · Program_Chairs · 2018-01-29
**ICLR 2018 Conference Acceptance Decision**

**Decision:**

Reject

**Comment:**

This paper is nowhere near standards for publication anywhere.